# Nested Counterfactual Identification
# from Arbitrary Surrogate Experiments

**Juan D. Correa**
Columbia University
jdcorrea@cs.columbia.edu

**Sanghack Lee**
Seoul National University
sanghack@snu.ac.kr

**Elias Bareinboim**
Columbia University
eb@cs.columbia.edu

## Abstract

The *Ladder of Causation* describes three qualitatively different types of activities an agent may be interested in engaging in, namely, seeing (observational), doing (interventional), and imagining (counterfactual) (Pearl and Mackenzie, 2018). The inferential challenge imposed by the causal hierarchy is that data is collected by an agent observing or intervening in a system (layers 1 and 2), while its goal may be to understand what would have happened had it taken a different course of action, contrary to what factually ended up happening (layer 3). While there exists a solid understanding of the conditions under which cross-layer inferences are allowed from observations to interventions, the results are somewhat scarcer when targeting counterfactual quantities. In this paper, we study the identification of nested counterfactuals from an arbitrary combination of observations and experiments. Specifically, building on a more explicit definition of nested counterfactuals, we prove the counterfactual unnesting theorem (CUT), which allows one to map arbitrary nested counterfactuals to unnested ones. For instance, applications in mediation and fairness analysis usually evoke notions of direct, indirect, and spurious effects, which naturally require nesting. Second, we introduce a sufficient and necessary graphical condition for counterfactual identification from an arbitrary combination of observational and experimental distributions. Lastly, we develop an efficient and complete algorithm for identifying nested counterfactuals; failure of the algorithm returning an expression for a query implies it is not identifiable.

## 1 Introduction

Counterfactuals provide the basis for notions pervasive throughout human affairs, such as credit assignment, blame and responsibility, and regret. One of the most powerful constructs in human reasoning —"what if?" questions— evokes hypothetical conditions usually contradicting the factual evidence. Judgment and understanding of critical situations found from medicine to psychology to business involve counterfactual reasoning, e.g.: "Joe received the treatment and died, would he be alive had he not received it?," "Had the candidate been male instead of female, would the decision from the admissions committee be more favorable?," or "Would the profit this quarter remain within 5% of its value had we increased the price by 2%?". By and large, counterfactuals are key ingredients that go in the construction of explanations about why things happened as they did [17, 19].

The structural interpretation of causality provides proper semantics for representing counterfactuals [17, Ch. 7]. Specifically, each structural causal model (SCM) $\mathcal{M}$ induces a collection of distributions related to the activities of seeing (called observational), doing (interventional), and imagining (counterfactual). The collection of these distributions is known as the *Ladder of causation* [19], and has also been called the *Pearl's Causal Hierarchy* (PCH, for short) [2]. The PCH is a containment hierarchy; each type of distribution can be put in increasingly refined layers: observational content goes in layer 1; experimental in layer 2; counterfactual in layer 3; see Fig. 1.

35th Conference on Neural Information Processing Systems (NeurIPS 2021).

It is understood that if we have all the information in the world about layer 1, there are still questions about layers 2 and 3 that are unanswerable, or technically undetermined; further, if we have data from layers 1 and 2, there are still questions in the world about layer 3 that are underdetermined [17, 2].

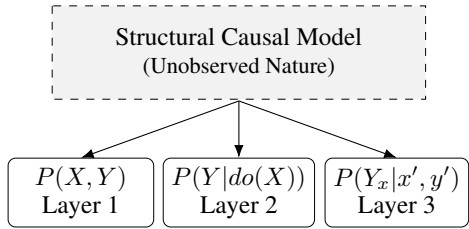

Figure 1: Every SCM induces different quantities in each layer of the PCH.

The inferential challenge in these settings arises because the generating model $\mathcal{M}$ is not fully observed, nor data from all of the layers are necessarily available, perhaps due to the cost or the infeasibility of performing certain interventions. One common task found in the literature is to determine the effect of an intervention of a variable $X$ on an outcome $Y$, say $P(Y|do(X))$ (layer 2), using data from observations $P(\mathbf{V})$ (layer 1), where $\mathbf{V}$ is the set of observed variables, and possibly other interventions, e.g., $P(\mathbf{V}|do(Z))$. Also, qualitative assumptions about the system are usually articulated in the form of a causal diagram $\mathcal{G}$. This setting has been studied in the literature under the rubric of non-parametric identification from a combination of observations and experiments. Multiple solutions exist, including Pearl's celebrated do-calculus [16], and other increasingly refined methods that are computationally efficient, sufficient, and necessary [25, 8, 20, 26, 21, 10, 3, 13, 12].[1]

There is a growing literature about cross-layer inferences from data in layers 1 and 2 to quantities in layer 3. For example, a data scientist may be interested in evaluating the effect of an intervention on the group of subjects who receive the treatment instead of those randomly assigned to it. This measure is known as the *effect of treatment on the treated* [9, 17], and there exists a graphical condition for mapping it to a (layer 2) causal effect [23]. Further, there are also results on the identification of *path-specific effects*, which are counterfactuals that isolate specific paths in the graph [18, 1]. In particular, [24] provides a sufficient and necessary algorithm for identification of these effects from observational data, and [28] provides identification conditions from observational and experimental data in general canonical models. Further, [22] studied counterfactual identification under the assumption that all experimental distributions (i.e., over every subset of the observed variables) are available.[2]

In this paper, our goal is to identify the probability distribution of (possibly nested) counterfactual events from an *arbitrary* combination of user-specified observational and experimental distributions. To the best of our knowledge, this provides the first general treatment of nested counterfactual identification from arbitrary data collections. Moreover, it also provides the first, graphical and algorithmic, sufficient and necessary conditions for the identifications of counterfactuals from observational data alone (when no experimental data is available) and arbitrary causal diagrams. Moving up the PCH, our results allow for arbitrary quantities as inferential targets and for the addition of arbitrary experimental distributions to the input, increasing the flexibility of the solution.

For concreteness, consider the causal diagram shown in Fig. 2 and a counterfactual query called *direct effect*. This quantity represents the sensitivity of a variable $Y$ to changes in another variable $X$ while all other factors in the analysis remain fixed. Suppose $X$ is level of exercise, $M$ cholesterol levels, and $Y$ cardiovascular disease. Exercising can improve cholesterol levels, which in turn affect the chances of developing cardiovascular disease. An interesting question is how much exercise prevents the disease by means other than regulating cholesterol. In counterfactual notation, this is to compare $Y_{x,M_x}$ and $Y_{x',M_x}$, where $x$

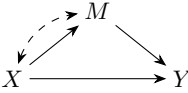

Figure 2: Causal diagram with treatment $X$, outcome $Y$, and mediator $M$.

and $x'$ are different values. The first quantity represents the value of $Y$ when $X=x$ and $M$ varies accordingly. The second expression is the value $Y$ attains if $X$ is held constant at $x'$ while $M$ still follows $X=x$. The difference $E[Y_{x',M_x} - Y_{x,M_x}]$ — known as the natural direct effect (NDE) — is non-zero if there is some direct effect of $X$ on $Y$. In this instance, this nested counterfactual is identifiable only if observational data and experiments on $X$ are available.

---

[1]In fact, this is a classic task in a larger family of problems known as data fusion, which include other challenges such as selection bias, transportability, to cite a few. For more details, see [4].

[2]For the sake of context, the work proposed here can be seen as a generalization of two tasks, counterfactual identification under the assumptions discussed earlier [22] and interventional identification from arbitrary experiments [13]. As discussed later on, we will be able to show, based on the machinery developed here, that the individual methods for those tasks can be combined and also be shown complete.

After all, there is no general identification method for this particular counterfactual family (which also includes indirect and spurious effects) and, more broadly, other arbitrary nested counterfactuals that are well-defined in layer 3. Our goal is to understand the non-parametric identification of arbitrary nested and conditional counterfactuals when the input consists of any combination of observational and interventional distributions, whatever is available for the data scientist. More specifically, our contributions are as follows.

1. We look at nested counterfactuals from an SCM perspective and introduce machinery that supports counterfactual reasoning. In particular, we prove the counterfactual unnesting theorem (CUT), which allows one to map any nested counterfactual to an unnested one (Section 2).
2. Building on this new machinery, we derive sufficient and necessary graphical conditions and an algorithm to determine the identifiability of marginal nested counterfactuals from an arbitrary combination of observational and experimental distributions (Section 3).
3. We prove a reduction from conditional counterfactuals to marginal ones, and use it to derive a complete algorithm for their identification (Section 4).

Due to space constraints, all the proofs in this paper can be found in the full technical report [5].

## 1.1 Preliminaries

We denote variables by capital letters, $X$, and values by small letters, $x$. Bold letters, $\mathbf{X}$ represent sets of variables and $\mathbf{x}$ sets of values. The domain of a variable $X$ is denoted by $\mathfrak{X}_X$. Two values $\mathbf{x}$ and $\mathbf{z}$ are said to be consistent if they share the common values for $\mathbf{X} \cap \mathbf{Z}$. We also denote by $\mathbf{x} \setminus \mathbf{Z}$ the value of $\mathbf{X} \setminus \mathbf{Z}$ consistent with $\mathbf{x}$ and by $\mathbf{x} \cap \mathbf{Z}$ the subset of $\mathbf{x}$ corresponding to variables in $\mathbf{Z}$. We assume the domain of every variable is finite.

We relay on causal graphs and denote them with a calligraphic letter, e.g., $\mathcal{G}$. We denote the set of vertices (i.e., variables) in $\mathcal{G}$ as $\mathbf{V}(\mathcal{G})$. Given a graph $\mathcal{G}$, $\mathcal{G}_{\overline{\mathbf{W}}\underline{\mathbf{X}}}$ is the result of removing edges coming into variables in $\mathbf{W}$ and going out from variables in $\mathbf{X}$. $\mathcal{G}[\mathbf{W}]$ denotes a vertex-induced subgraph including $\mathbf{W}$ and the edges among its elements. We use kinship notation for graphical relationships such as parents, children, descendants, and ancestors of a set of variables. For example, the set of parents of $\mathbf{X}$ in $\mathcal{G}$ is $Pa(\mathbf{X})_\mathcal{G} := \mathbf{X} \cup \bigcup_{X \in \mathbf{X}} Pa(X)_\mathcal{G}$. Similarly, we define $Ch()$, $De()$, and $An()$.

To articulate and formalize counterfactual questions, we require a framework that allows us to reason simultaneously about events from *alternative worlds*. Accordingly, we employ the Structural Causal Model (SCM) paradigm [17, Ch. 7]. An SCM $\mathcal{M}$ is a 4-tuple $\langle \mathbf{U}, \mathbf{V}, \mathcal{F}, P(\mathbf{u}) \rangle$, where $\mathbf{U}$ is a set of exogenous (latent) variables; $\mathbf{V}$ is a set of endogenous (observable) variables; $\mathcal{F}$ is a collection of functions such that each variable $V_i \in \mathbf{V}$ is determined by a function $f_i \in \mathcal{F}$. Each $f_i$ is a mapping from a set of exogenous variables $\mathbf{U}_i \subseteq \mathbf{U}$ and a set of endogenous variables $\mathbf{Pa}_i \subseteq \mathbf{V} \setminus \{V_i\}$ to the domain of $V_i$. Uncertainty is encoded through a probability distribution over the exogenous variables, $P(\mathbf{U})$. An SCM $\mathcal{M}$ induces a *causal diagram* $\mathcal{G}$ where $\mathbf{V}$ is the set of vertices, there is a directed edge $(V_j \rightarrow V_i)$ for every $V_i \in \mathbf{V}$ and $V_j \in \mathbf{Pa}_i$, and a bidirected edge $(V_i \leftarrow\!\!-\!\!\rightarrow V_j)$ for every pair $V_i, V_j \in \mathbf{V}$ such that $U_i \cap U_j \neq \emptyset$ ($V_i$ and $V_j$ have a common exogenous parent).

We assume that the underlying model is recursive. That is, there are no cyclic dependencies among the variables. Equivalently, that is to say, that the corresponding causal diagram is acyclic. The set $\mathbf{V}(\mathcal{G})$ can be partitioned into subsets called *c-components* [27] such that two variables belong to the same c-component if they are connected in $\mathcal{G}$ by a path made entirely of bidirected edges.

## 2 SCMs and Nested Counterfactuals

Intervening on a system represented by an SCM $\mathcal{M}$ results in a new model differing only on the mechanisms associated with the intervened variables [15, 6, 7]. If the intervention consists on fixing the value of a variable $X$ to a constant $x \in \mathfrak{X}_X$, it induces a *submodel*, denoted $\mathcal{M}_x$ [17, Def. 7.1.2]. To formally study nested counterfactuals, we extend this notion to models derived from interventions that replace functions from the original SCM with other, not necessarily constant, functions.

**Definition 1** (Derived Model). Let $\mathcal{M}$ be an SCM, $\widehat{\mathbf{U}} \subseteq \mathbf{U}$, $X \in \mathbf{V}$, and $\widehat{X} : \widehat{\mathbf{U}} \rightarrow \mathfrak{X}_X$ a function. Then, $\mathcal{M}_{\widehat{X}}$, called the *derived model* of $\mathcal{M}$ according to $\widehat{X}$, is identical to $\mathcal{M}$, except that the function $f_X$ is replaced with a function $\widehat{f}_X$ identical to $\widehat{X}$.

This definition is easily extendable to models derived from an intervention on a set $\mathbf{X}$ instead of a singleton. When $\widehat{\mathbf{X}}$ is a collection of functions $\{\widehat{X} : \widehat{\mathbf{U}}_X \to \mathfrak{X}_X\}_{X \in \mathbf{X}}$, the derived model $\mathcal{M}_{\widehat{\mathbf{X}}}$ is obtained by replacing each $f_X$ with $\widehat{X}$ for $X \in \mathbf{X}$. Next, we discuss the concept of *potential response* [17, Def. 7.4.1] with respect to the derived models.

**Definition 2** (Potential Response). Let $\mathbf{X}, \mathbf{Y} \subseteq \mathbf{V}$ be subsets of observable variables, let $\mathbf{u}$ be a unit, and let $\widehat{\mathbf{X}}(\mathbf{u})$ be a set of functions from $\widehat{\mathbf{U}}_X \to \mathfrak{X}_X$, for $X \in \mathbf{X}$ where $\widehat{\mathbf{U}}_X \subseteq \mathbf{U}$. Then, $\mathbf{Y}_{\mathbf{X}=\widehat{\mathbf{X}}}(\mathbf{u})$ (or $\mathbf{Y}_{\widehat{\mathbf{X}}}(\mathbf{u})$, for short) is called the *potential response* of $\mathbf{Y}$ to $\mathbf{X} = \widehat{\mathbf{X}}$, and is defined as the solution of $\mathbf{Y}$, for a particular $\mathbf{u}$, in the derived model $\mathcal{M}_{\widehat{\mathbf{X}}}$.

A potential response $Y_{\widehat{\mathbf{X}}}(\mathbf{u})$ describes the value that variable $Y$ would attain for a unit (or individual) $\mathbf{u}$ if the intervention $\widehat{\mathbf{X}}$ is performed. This concept is tightly related to that of *potential outcome*, but the former explicitly allows for interventions that do not necessarily fix the variables in $\mathbf{X}$ to a constant value. Averaging over the space of $\mathbf{U}$, a potential response $Y_{\widehat{\mathbf{X}}}(\mathbf{u})$ induces a random variable that we will denote simply as $Y_{\widehat{\mathbf{X}}}$. If the intervention replaces a function $f_X$ with a potential response of $X$ in $\mathcal{M}$, we say the intervention is *natural*.

When variables are enumerated as $W_1, W_2, \ldots$, we may add square brackets around the part of the subscript denoting interventions. We use $\mathbf{W}_*$ to denote sets of arbitrary counterfactual variables. Let $\mathbf{W}_* = \{W_{1[\widehat{\mathbf{T}}_1]}, W_{2[\widehat{\mathbf{T}}_2]}, \ldots\}$ represent a set of counterfactual variables such that $W_i \in \mathbf{V}$ and $\mathbf{T}_i \subseteq \mathbf{V}$ for $i = 1, \ldots, l$. Define $\mathbf{V}(\mathbf{W}_*) = \{W \in \mathbf{V} \mid W_{\widehat{\mathbf{T}}} \in \mathbf{W}_*\}$, that is, the set of observables that appear in $\mathbf{W}_*$. Let $\mathbf{w}_*$ represent a vector of values, one for each variable in $\mathbf{W}_*$ and define $\mathbf{w}_*(\mathbf{X}_*)$ as the subset of $\mathbf{w}_*$ corresponding to $\mathbf{X}_*$ for any $\mathbf{X}_* \subseteq \mathbf{W}_*$.

The probability of any counterfactual event is given by

$$P(\mathbf{Y}_* = \mathbf{y}_*) = \sum\nolimits_{\{\mathbf{u} | \mathbf{Y}_*(\mathbf{u}) = \mathbf{y}_*\}} P(\mathbf{u}), \tag{1}$$

where the predicate $\mathbf{Y}_*(\mathbf{u}) = \mathbf{y}_*$ means $\bigwedge_{\{Y_{\widehat{\mathbf{X}}} \in \mathbf{Y}_*\}} Y_{\widehat{\mathbf{X}}}(\mathbf{u}) = y$.

When all variables in the expression have the same subscript, that is, they belong to the same submodel; we will often denote it as $P_{\mathbf{x}}(W_1, W_2, \ldots)$.

For most real-world scenarios, having access to a fully specified SCM of the underlying system is unfeasible. Nevertheless, our analysis does not rely on such privileged access but the aspects of the model captured by the causal graph and data samples generated by the unobserved model.

## 2.1 Nested Counterfactuals

Potential responses can be compounded based on natural interventions. For instance, the counterfactual $Y_{Z_x}(\mathbf{u})$ ($Y_{Z=Z_x}(\mathbf{u})$) can be seen as the potential response of $Y$ to an intervention that makes $\widehat{Z}$ equal to $Z_x$. Notice that $Z_x(\mathbf{u})$ is in itself a potential response, but from a different (nested) model. Hence we call $Y_{Z_x}$ a *nested counterfactual*.

Recall the causal diagram in Fig. 2 and consider once again the NDE as

$$NDE_{x \to x', Z}(Y) = E[Y_{x' Z_x}] - E[Y_x]. \tag{2}$$

The second term is also equal to $Y_{x Z_x}$ as $Z_x$ is consistent with $X = x$, so it is the value $Y$ listens to in $\mathcal{M}_x$. Meanwhile, the first one is related to $P(Y_{x' Z_x})$, the probability of a nested counterfactual.

## 2.2 Tools for Counterfactual Reasoning

Before characterizing the identification of counterfactuals from observational and experimental data, we develop from first principles a canonical representation of any such query. First, we extend the notion of ancestors for counterfactual variables, which subsumes the usual one described before.

**Definition 3** (Ancestors, of a counterfactual). Let $Y_{\mathbf{x}}$ be such that $Y \in \mathbf{V}, \mathbf{X} \subseteq \mathbf{V}$. Then, the set of (counterfactual) ancestors of $Y_{\mathbf{x}}$, denoted $An(Y_{\mathbf{x}})$, consist of each $W_{\mathbf{z}}$, such that $W \in An(Y)_{\mathcal{G}_{\underline{\mathbf{X}}}}$ (which includes $Y$ itself), and $\mathbf{z} = \mathbf{x} \cap An(W)_{\mathcal{G}_{\overline{\mathbf{X}}}}$.

For a set of variables $\mathbf{W}_*$, we define $An(\mathbf{W}_*)$ as the union of the ancestors of each variable in the set. That is, $An(\mathbf{W}_*) = \bigcup_{W_{\mathbf{t}} \in \mathbf{W}_*} An(W_{\mathbf{t}})$. For instance, in Fig. 3(a), $An(Y_x) = \{Y_x, Z\}$, $An(X_{yz}) =$

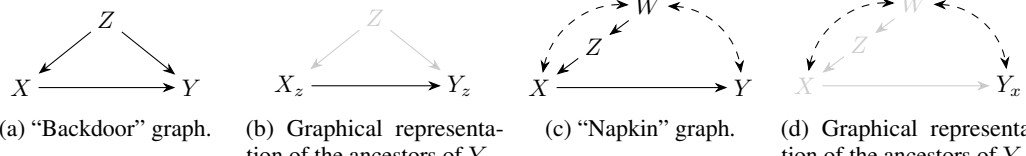

(a) "Backdoor" graph.  (b) Graphical representation of the ancestors of $Y_z$.  (c) "Napkin" graph.  (d) Graphical representation of the ancestors of $Y_x$.

Figure 3: Two causal diagrams and the subgraphs considered when finding sets of ancestors for a counterfactual variable.

$\{X_z\}$ and $An(Y_z) = \{Y_z, X_z\}$ (depicted in Fig. 3(b)). In Fig. 3(c) $An(Z, Y_z) = \{Y_z, X_z, Z, W\}$ and $An(Y_x) = \{Y_x\}$ (represented in Fig. 3(d)).

Probabilistic and causal inference with graphical models exploits the local structure among variables, specifically parent-child relationships, to infer and even estimate probabilities. In particular, Tian [27] introduced *c-factors* which have proven instrumental in solving many problems in causal inference. We naturally generalize this notion to the counterfactual setting with the following definition.

**Definition 4** (Counterfactual Factor (ctf-factor)). A counterfactual factor is a distribution of the form

$$P(W_{1[\mathbf{pa}_1]} = w_1, W_{2[\mathbf{pa}_2]} = w_2, \ldots, W_{l[\mathbf{pa}_l]} = w_l), \tag{3}$$

where each $W_i \in \mathbf{V}$ and there could be $W_i = W_j$ for some $i, j \in \{1, \ldots, l\}$.

For example, for Fig. 3(c) $P(Y_x = y, Y_{x'} = y')$, $P(Y_x = y, X_z = x)$ are ctf-factors but $P(Y_z = y, Z_w = z)$ is not. Using the notion of ancestrality introduced in Definition 3, we can factorize counterfactual probabilities as ctf-factors.

**Theorem 1** (Ancestral set factorization). *Let $\mathbf{W}_*$ be an ancestral set, that is, $An(\mathbf{W}_*) = \mathbf{W}_*$, and let $\mathbf{w}_*$ be a vector with a value for each variable in $\mathbf{W}_*$. Then,*

$$P(\mathbf{W}_* = \mathbf{w}_*) = P\left(\bigwedge_{W_{\mathbf{t}} \in \mathbf{W}_*} W_{\mathbf{pa}_w} = w\right), \tag{4}$$

*where each $w$ is $w_t$ and $\mathbf{pa}_w$ is determined for each $W_{\mathbf{t}} \in \mathbf{W}_*$ as follows:*

 (i)  *the values for variables in $\mathbf{Pa}_w \cap \mathbf{T}$ are the same as in $\mathbf{t}$, and*
 (ii) *the values for variables in $\mathbf{Pa}_w \setminus \mathbf{T}$ are taken from $\mathbf{w}_*$ corresponding to the parents of $W$.*

*Proof outline.* Following a reverse topological order in $\mathcal{G}$, look at each $W_{i\mathbf{t}_i} \in \mathbf{W}_*$. Since any parent of $W_i$ not in $\mathbf{T}_i$ must appear in $\mathbf{W}_*$, the composition axiom [17, 7.3.1] licenses adding them to the subscript. Then, by exclusion restrictions [16], any intervention not involving $Pa(W_i)$ can be removed to obtain the form in Eq. (4). □

For example, consider the diagram in Fig. 3(c) and the counterfactual $P(Y_x = y \mid X = x')$ known as the *effect of the treatment on the treated* (ETT) [9, 17]. First note that $P(Y_x = y \mid X = x') = P(Y_x = y, X = x')/P(X = x')$ and that $An(Y_x, X) = \{Y_x, X, Z, W\}$, then

$$P(Y_x = y, X = x') = \sum_{z,w} P(Y_x = y, X = x', Z = z, W = w). \tag{5}$$

Then, by Theorem 1 we can write

$$P(Y_x = y, X = x') = \sum_{z,w} P(Y_x = y, X_z = x', Z_w = z, W = w). \tag{6}$$

Moreover, the following result describes a factorization of ctf-factors based on the c-component structure of the graph, which will prove instrumental in the next section.

**Theorem 2** (Counterfactual factorization). *Let $P(\mathbf{W}_* = \mathbf{w}_*)$ be a ctf-factor, let $W_1 < W_2 < \cdots$ be a topological order over the variables in $\mathcal{G}[\mathbf{V}(\mathbf{W}_*)]$, and let $\mathbf{C}_1, \ldots, \mathbf{C}_k$ be the c-components of the same graph. Define $\mathbf{C}_{j*} = \{W_{\mathbf{pa}_w} \in \mathbf{W}_* \mid W \in \mathbf{C}_j\}$ and $\mathbf{c}_{j*}$ as the values in $\mathbf{w}_*$ corresponding to $\mathbf{C}_{j*}$, then $P(\mathbf{W}_* = \mathbf{w}_*)$ decomposes as*

$$P(\mathbf{W}_* = \mathbf{w}_*) = \prod_j P(\mathbf{C}_{j*} = \mathbf{c}_{j*}). \tag{7}$$

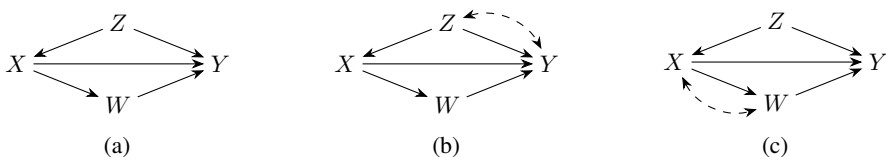

(a)                  (b)                  (c)

Figure 4: Three causal diagrams representing plausible structures in mediation analysis.

*Furthermore, each factor can be computed from $P(\mathbf{W}_* = \mathbf{w})$ as*

$$P(\mathbf{C}_{j*} = \mathbf{c}_{j*}) = \prod_{\{W_i \in \mathbf{C}_j\}} \frac{\sum_{\{w | W_{\mathbf{pa}_w} \in \mathbf{W}_*, W_i < W\}} P(\mathbf{W}_* = \mathbf{w}_*)}{\sum_{\{w | W_{\mathbf{pa}_w} \in \mathbf{W}_*, W_{i-1} < W\}} P(\mathbf{W}_* = \mathbf{w}_*)}. \tag{8}$$

Armed with these results, we consider the identification problem in the next section.

## 3 Counterfactual Identification from Observations and Experiments

In this section, we consider the identification of a counterfactual probability from a collection of observational and experimental distributions. This task can be seen as a generalization of that in [13] where the available data is the same, but the query is a causal effect $P_{\mathbf{x}}(\mathbf{Y})$. Let $\mathbb{Z} = \{\mathbf{Z}_1, \mathbf{Z}_2, \ldots\}, \mathbf{Z}_j \subseteq \mathbf{V}$, and assume that all of $\{P_{\mathbf{z}_j}(\mathbf{V})\}_{\mathbf{z}_j \in \mathfrak{X}_{\mathbf{Z}_j}, \mathbf{Z}_j \in \mathbb{Z}}$ are available. Notice that $\mathbf{Z}_j = \emptyset$ is a valid choice corresponding to $P(\mathbf{V})$ the observational (non-interventional) distribution.

**Definition 5** (Counterfactual Identification). A query $P(\mathbf{Y}_* = \mathbf{y}_*)$ is said to be identifiable from $\mathbb{Z}$ in $\mathcal{G}$, if $P(\mathbf{Y}_* = \mathbf{y}_*)$ is uniquely computable from the distributions $\{P_{\mathbf{z}_j}(\mathbf{V})\}_{\mathbf{z}_j \in \mathfrak{X}_{\mathbf{Z}_j}, \mathbf{Z}_j \in \mathbb{Z}}$ in any causal model which induces $\mathcal{G}$.

Given an arbitrary query $P(\mathbf{Y}_* = \mathbf{y}_*)$, we could express it in terms of ctf-factors by writing $P(\mathbf{Y}_* = \mathbf{y}_*) = \sum_{\mathbf{d}_* \setminus \mathbf{y}_*} P(\mathbf{D}_* = \mathbf{d}_*)$ where $\mathbf{D}_* = An(\mathbf{Y}_*)$ and then using Theorem 1 to write $P(\mathbf{D}_* = \mathbf{d}_*)$ as a ctf-factor. For instance, the ancestral set $\mathbf{W}_* = \{Y_x, X, Z, W\}$ with $\mathbf{w} = \{y, x', z, w\}$ in Eq. (6) can be written in terms of ctf-factors as

$$P(Y_x = y, X_z = x', Z_w = z, W = w) = P(Y_x = y, X_z = x', W = w)P(Z_w = z). \tag{9}$$

The following lemma characterizes the relationship between the identifiability of $P(\mathbf{Y}_* = \mathbf{y}_*)$ and $P(\mathbf{D}_* = \mathbf{d}_*)$.

**Lemma 1.** *Let $P(\mathbf{W}_* = \mathbf{w}_*)$ be a ctf-factor and let $\mathbf{Y}_* \subseteq \mathbf{W}_*$ be such that $\mathbf{W}_* = An(\mathbf{Y}_*)$. Then, $\sum_{\mathbf{w}_* \setminus \mathbf{y}_*} P(\mathbf{W}_* = \mathbf{w}_*)$ is identifiable from $\mathbb{Z}$ if and only if $P(\mathbf{W}_* = \mathbf{w}_*)$ is identifiable from $\mathbb{Z}$.*

Once the query of interest is in ctf-factor-form, the identification task reduces to identifying smaller ctf-factors according to the c-components of $\mathcal{G}$. In this respect, Theorem 2 implies the following

**Corollary 1.** *Let $P(\mathbf{W}_* = \mathbf{w}_*)$ be a ctf-factor and $\mathbf{C}_j$ be a c-component of $\mathcal{G}[\mathbf{V}(\mathbf{W}_*)]$. Then, if $P(\mathbf{C}_{j*} = \mathbf{c}_{j*})$ is not identifiable, $P(\mathbf{W}_* = \mathbf{w}_*)$ is also not identifiable.*

*Proof.* Assume for the sake of contradiction that $P(\mathbf{C}_{j*} = \mathbf{c}_{j*})$ is not identifiable but $P(\mathbf{W}_* = \mathbf{w}_*)$ is. Then, by Theorem 2, the former is identifiable from the latter, a contradiction. $\square$

Let us consider the causal diagrams in Fig. 4 and the counterfactual $Y_{x_1, W_{x_0}} = y, X = x$, with $x_0, x_1, x \in \mathfrak{X}_X$, used to define quantities for fairness analysis in [28] (e.g., $Y_{x_1, W_{x_0}} = y | X = x$):

$$P(Y_{x_1, W_{x_0}} = y, X = x)$$

$$= \sum_w P(Y_{x_1, w} = y, W_{x_0} = w, X = x) \qquad \text{Unnesting} \tag{10}$$

$$= \sum_{w,z} P(Y_{x_1, w} = y, W_{x_0} = w, X = x, Z = z) \qquad \text{Complete ancestral set} \tag{11}$$

$$= \sum_{w,z} P(Y_{x_1, w, z} = y, W_{x_0} = w, X_z = x, Z = z) \qquad \text{Write in ctf-factor-form} \tag{12}$$

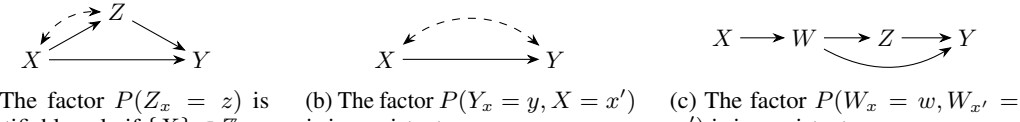

(a) The factor $P(Z_x = z)$ is identifiable only if $\{X\} \in \mathbb{Z}$.

(b) The factor $P(Y_x = y, X = x')$ is inconsistent.

(c) The factor $P(W_x = w, W_{x'} = w')$ is inconsistent.

Figure 5: Examples of causal diagrams and inconsistent ctf-factors derived from them.

Due to the particular c-component structure of each model, we can factorize $P(Y_{x_1,w,z} = y, W_{x_0} = w, X_z = x, Z = z)$ according to each model as:

$$P(Y_{x_1,w,z} = y)P(W_{x_0} = w)P(X_z = x)P(Z = z), \tag{13}$$

$$P(Y_{x_1,w,z} = y, Z = z)P(W_{x_0} = w)P(X_z = x), \text{ and} \tag{14}$$

$$P(Y_{x_1,w,z} = y)P(W_{x_0} = w, X_z = x)P(Z = z). \tag{15}$$

The question then becomes, whether ctf-factors corresponding to individual c-components can be identified from the available input. In this example, all factors in Eq. (13) and Eq. (14) are identifiable from $P(\mathbf{V})$. For Eq. (14) in particular, they are given by

$$P(Y = y, Z = z \mid W = w, X = x_1)P(W = w \mid X = x_0)P(X = x \mid Z = z). \tag{16}$$

In contrast, the factor $P(W_{x_0}=w, X_z=x)$ in Eq. (15) (model Fig. 4(c)) is only identifiable if $x=x_0$. The following definition and theorem characterize the factors that can be identified from $\mathbb{Z}$ and $\mathcal{G}$.

**Definition 6** (Inconsistent ctf-factor). $P(\mathbf{W}_* = \mathbf{w}_*)$ is an inconsistent ctf-factor if it is a ctf-factor, $\mathcal{G}[\mathbf{V}(\mathbf{W}_*)]$ has a single c-component, and one of the following situations hold:

(i) there exist $W_{\mathbf{t}} \in \mathbf{W}_*, Z \in \mathbf{T} \cap \mathbf{V}(\mathbf{W}_*)$ such that $z \in \mathbf{t}, z' \in \mathbf{w}_*$ and $z \neq z'$, or

(ii) there exists $W_{i[\mathbf{t}_i]}, W_{j[\mathbf{t}_j]} \in \mathbf{W}_*$ and $T \in \mathbf{T}_i \cap \mathbf{T}_j$ such that $t \in \mathbf{t}_1, t' \in \mathbf{t}_2$ and $t \neq t'$.

**Theorem 3** (Ctf-factor identifiability). *A ctf-factor $P(\mathbf{W}_* = \mathbf{w})$ is identifiable from $\mathbb{Z}$ if and only if it is consistent. If consistent, let $\mathbf{W} = \mathbf{V}(\mathbf{W}_*)$ and $\mathbf{W}' = \mathbf{V} \setminus \mathbf{W}$; then $P(\mathbf{W}_* = \mathbf{w}_*)$ is equal to $P_{\mathbf{w}'}(\mathbf{w})$ where $\mathbf{w}$ and $\mathbf{w}'$ are consistent with $\mathbf{w}_* \cup \bigcup_{\{W_{\mathbf{pa}_w} \in \mathbf{W}_*\}} \mathbf{pa}_w$.*

Consider the $NDE_{x \to x', Z}(Y)$ in Fig. 5(a), we can write

$$P(Y_{x'Z_x} = y) = \sum_z P(Y_{x'z} = y, Z_x = z) = \sum_z P(Y_{x'z} = y)P(Z_x = z). \tag{17}$$

While the factor $P(Y_{x'z} = y)$ is identifiable from $P(\mathbf{V})$ as $P(Y = y \mid X = x', Z = z)$, the second factor is identifiable only if experimental data on $X$ is available, as $P_x(z)$.

We can also verify that the factor $P(Y_x = y, X = x')$ in Fig. 5(b) is inconsistent. For another example consider the ETT-like expression $P(Y_{x,z} = y, X = x', Z = z')$ in Fig. 5(c), we have

$$P(Y_{xz} = y, X = x', Z = z')$$

$$= \sum_{w,w'} P(Y_{xz} = y, X = x', Z = z', W_x = w, W = w') \tag{18}$$

$$= \sum_{w,w'} P(Y_{xz} = y, X = x', Z_{w'} = z', W_x = w, W_{x'} = w') \tag{19}$$

$$= \sum_{w,w'} P(Y_{xz} = y)P(X = x')P(Z_{w'} = z')P(W_x = w, W_{x'} = w'), \tag{20}$$

where the factor $P(W_x = w, W_{x'} = w')$ is inconsistent.

Given a counterfactual variable $\mathbf{Y}_{\mathbf{x}}$, it could be the case that some values in $\mathbf{x}$ become causally irrelevant to $Y$ after the rest of $\mathbf{x}$ has been fixed. Formally,

**Lemma 2** (Interventional Minimization). *Let $\|\mathbf{Y}_*\| := \bigcup_{Y_{\widehat{\mathbf{X}}} \in \mathbf{Y}_*} \|Y_{\widehat{\mathbf{X}}}\|$ such that $\|Y_{\widehat{\mathbf{X}}}\| := Y_{\widehat{\mathbf{Z}}}$ where $\mathbf{Z} = \mathbf{X} \cap An(Y)_{\mathcal{G}_{\overline{\mathbf{X}}}}$ and $\widehat{\mathbf{Z}}$ is consistent with $\|\widehat{\mathbf{X}}\|$, $\|\mathbf{x}\| = \mathbf{x}$ and $\|\emptyset\| = \emptyset$. Then, $\mathbf{Y}_* = \|\mathbf{Y}_*\|$.*

Moreover, such simplification may reveal counterfactual expressions with equivalent or contradicting events. In Fig. 3(c), $(Y_{xz} = y, Y_{xz'} = y') = (Y_x = y, Y_x = y')$ which has probability 0 if $y \neq y'$, or $(Y_{xz} = y, Y_{xz'} = y)$ that is simply $(Y_x = y)$. Similarly, the probabilities of counterfactual events of the form $P(X_x = x'), x \neq x'$, and $P(X_x = x)$ are trivially 0 and 1 respectively.

The following result shows how nested counterfactuals can be written in terms of non-nested ones.

**Algorithm 1** CTFIDU($\mathbf{Y}_*, \mathbf{y}_*, \mathbb{Z}, \mathcal{G}$)

**Input**: $\mathcal{G}$ causal diagram over variables $\mathbf{V}$; $\mathbf{Y}_*$ a set of counterfactual variables in $\mathbf{V}$; $\mathbf{y}_*$ a set of values for $\mathbf{Y}_*$; and available distribution specification $\mathbb{Z}$.
**Output**: $P(\mathbf{Y}_* = \mathbf{y}_*)$ in terms of available distributions or FAIL if not identifiable from $\langle \mathcal{G}, \mathbb{Z} \rangle$.

1: let $\mathbf{Y}_* \leftarrow \|\mathbf{Y}_*\|$.
2: **if** there exists $Y_{\mathbf{x}} \in \mathbf{Y}_*$ with two or more different values in $\mathbf{y}_*(Y_{\mathbf{x}})$ or $Y_y \in \mathbf{Y}_*$ with $\mathbf{y}_*(Y_y) \neq y$ **then return** 0.
3: **if** there exists $Y_{\mathbf{x}} \in \mathbf{Y}_*$ with two consistent values in $\mathbf{y}_*(Y_{\mathbf{x}})$ or $Y_y \in \mathbf{Y}_*$ with $\mathbf{y}_*(Y_y) = y$ **then** remove repeated variables from $\mathbf{Y}_*$ and values $\mathbf{y}_*$.
4: let $\mathbf{W}_* \leftarrow An(\mathbf{Y}_*)$, and let $\mathbf{C}_{1*}, \ldots, \mathbf{C}_{k*}$ be corresponding ctf-factors in $\mathcal{G}[\mathbf{V}(\mathbf{W}_*)]$.
5: **for each** $\mathbf{C}_i$ s.t. $(\mathbf{C}_{i*} = \mathbf{c}_{i*})$ is not inconsistent, $\mathbf{Z} \in \mathbb{Z}$ s.t. $\mathbf{C}_i \cap \mathbf{Z} = \emptyset$ **do**
6:     let $\mathbf{B}_i$ be the c-component of $\mathcal{G}_{\overline{\mathbf{Z}}}$ such that $\mathbf{C}_i \subseteq \mathbf{B}_i$, compute $P_{\mathbf{V} \backslash \mathbf{B}_i}(\mathbf{B}_i)$ from $P_{\mathbf{Z}}(\mathbf{V})$.
7:     **if** IDENTIFY$(\mathbf{C}_i, \mathbf{B}_i, P_{\mathbf{V} \backslash \mathbf{B}_i}(\mathbf{B}_i), \mathcal{G})$ does not FAIL **then**
8:         let $P_{\mathbf{V} \backslash \mathbf{C}_i}(\mathbf{C}_i) \leftarrow$ IDENTIFY$(\mathbf{C}_i, \mathbf{B}_i, P_{\mathbf{V} \backslash \mathbf{B}_i}(\mathbf{B}_i), \mathcal{G})$.
9:         let $P(\mathbf{C}_{i*} = \mathbf{c}_{i*}) \leftarrow P_{\mathbf{V} \backslash \mathbf{C}_i}(\mathbf{C}_i)$ evaluated with values $(\mathbf{c}_{i*} \cup \bigcup_{C_{\mathbf{t}} \in \mathbf{C}_{i*}} \mathbf{pa}_c)$.
10:        move to the next $\mathbf{C}_i$.
11:    **end if**
12: **end for**
13: **if** any $P(\mathbf{C}_{i*} = \mathbf{c}_{i*})$ is inconsistent or was not identified from $\mathbb{Z}$ **then return** FAIL.
14: **return** $P(\mathbf{Y}_* = \mathbf{y}_*) \leftarrow \sum_{\mathbf{w}_* \backslash \mathbf{y}_*} \prod_i P(\mathbf{C}_{i*} = \mathbf{c}_{i*})$.

**Theorem 4** (Counterfactual Unnesting Theorem (CUT)). *Let $\widehat{\mathbf{X}}, \widehat{\mathbf{Z}}$ be any natural interventions on disjoint sets $\mathbf{X}, \mathbf{Z} \subseteq \mathbf{V}$. Then, for $\mathbf{Y} \subseteq \mathbf{V}$ disjoint from $\mathbf{X}$ and $\mathbf{Z}$ such that $\mathbf{X} \subseteq An(\mathbf{Y})_{\mathcal{G}_{\overline{\mathbf{Z}}}}$, $P(\mathbf{Y}_{\widehat{\mathbf{Z}}, \widehat{\mathbf{X}}} = \mathbf{y})$ is identifiable iff $P(\mathbf{Y}_{\widehat{\mathbf{Z}}, \mathbf{x}} = \mathbf{y}, \widehat{\mathbf{X}} = \mathbf{x})$ is identifiable for every $\mathbf{x}$, and given by*

$$P(\mathbf{Y}_{\widehat{\mathbf{Z}}, \widehat{\mathbf{X}}} = \mathbf{y}) = \sum_{\mathbf{x} \in \mathfrak{X}_{\mathbf{X}}} P(\mathbf{Y}_{\widehat{\mathbf{Z}}, \mathbf{x}} = \mathbf{y}, \widehat{\mathbf{X}} = \mathbf{x}). \tag{21}$$

For instance, for the model in Fig. 2 we can write

$$P(Y_{x' Z_x} = y) = \sum_z P(Y_{x' z} = y, Z_x = z). \tag{22}$$

As Theorem 4 allows us to rewrite any nested counterfactual in terms of a non-nested one, we focus on the latter and assume that any given counterfactual is already unnested.

Using the results in this section, we propose the algorithm CTFIDU (Algorithm 1) which given a set of counterfactual variables $\mathbf{Y}_*$, values $\mathbf{y}_*$, a collection of observational and experimental distributions $\mathbb{Z}$, and a causal diagram $\mathcal{G}$; outputs an expression for $P(\mathbf{Y}_* = \mathbf{y}_*)$ in terms of the specified distributions or FAIL if the query is not identifiable from such input in $\mathcal{G}$. Line 1 removes irrelevant subscripts from the query by virtue of Lemma 2. Then, lines 2 and 3 look for inconsistent events and redundant events, respectively. Line 4 finds the relevant ctf-factors consisting of a single c-component, as licensed by Theorem 1 and Theorem 2. As long as the factors are consistent, and allowed by Theorem 3, lines 6-11 carry out identification of the causal effect $P_{\mathbf{V} \backslash \mathbf{C}_i}(\mathbf{C}_i)$ from the available distributions employing the algorithm IDENTIFY [26] as a subroutine (see example in the next section). The procedure fails if any of the factors $P(\mathbf{C}_{i*} = \mathbf{c}_{i*})$ is inconsistent or not identifiable from $\mathbb{Z}$. Otherwise, it returns the corresponding expression.

**Theorem 5** (CTFIDU completeness). *A counterfactual probability $P(\mathbf{Y}_* = \mathbf{y}_*)$ is identifiable from $\mathbb{Z}$ and $\mathcal{G}$ if and only if CTFIDU returns an expression for it.*

This result ascertains that Algorithm 1 solves the identification task for counterfactuals in the form $P(\mathbf{Y}_* = \mathbf{y}_*)$.[3] Still, there exist other quantities that take into account evidence from events observed in the system, and cannot be written in this particular form. Since these queries represent important aspects of the underlying system, we will discuss them in the next section.

---

[3]One corollary of this result is that two identification algorithms ([22, 13]) can be combined and be shown complete; for details, see Appendix F. Even though our results were developed from first principles using an entirely new approach and machinery (which was the key to derive Algorithm 1 and prove its completeness), practitioners familiar with these algorithms may find the completeness of this combined approach attractive.

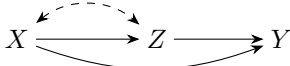

(a) While $P(Y_x=y \mid Z_x=z, X=x')$ is identifiable from $\mathbb{Z}$, $P(Y_x=y, Z_x=z, X=x')$ is not.

(b) $P(Y_x=y \mid Z_x=z, X=x')$ is not identifiable from $\mathbb{Z}$ because of the factor $P(Y_{xz}=y, X=x')$.

Figure 6: Examples of conditional queries.

---

**Algorithm 2** CTFID$(\mathbf{Y}_*, \mathbf{y}_*, \mathbf{X}_*, \mathbf{x}_*, \mathbb{Z}, \mathcal{G})$

---

**Input**: $\mathcal{G}$ causal diagram over variables $\mathbf{V}$; $\mathbf{Y}_*, \mathbf{X}_*$ a set of counterfactual variables in $\mathbf{V}$; $\mathbf{y}_*, \mathbf{x}_*$ a set of values for $\mathbf{Y}_*$ and $\mathbf{X}_*$; and available distribution specification $\mathbb{Z}$.
**Output**: $P(\mathbf{Y}_*=\mathbf{y}_* \mid \mathbf{X}_*=\mathbf{x}_*)$ in terms of available distributions or FAIL if non-ID from $\langle \mathcal{G}, \mathbb{Z} \rangle$.
 1: Let $\mathbf{A}_{1*}, \mathbf{A}_{2*}, \ldots$ be the ancestral components of $\mathbf{Y}_* \cup \mathbf{X}_*$ given $\mathbf{X}_*$.
 2: Let $\mathbf{D}_*$ be the union of the ancestral components containing a variable in $\mathbf{Y}_*$ and $\mathbf{d}_*$ the corresponding set of values.
 3: let $Q \leftarrow$ CTFIDU$(\bigcup_{D_\mathbf{t} \in \mathbf{D}_*} \mathbf{D}_{\mathbf{pa}_d}, \mathbf{d}_*, \mathbb{Z}, \mathcal{G})$.
 4: **return** $\sum_{\mathbf{d}_* \setminus (\mathbf{y}_* \cup \mathbf{x}_*)} Q / \sum_{\mathbf{d}_* \setminus \mathbf{x}_*} Q$.

---

## 4  Identification of Conditional Counterfactuals

In this section, we consider counterfactual quantities of the form $P(\mathbf{Y}_* = \mathbf{y}_* \mid \mathbf{X}_* = \mathbf{x}_*)$. It is immediate to write such a query as $P(\mathbf{W}_* = \mathbf{w}_*)/\sum_{\mathbf{y}_*} P(\mathbf{W}_* = \mathbf{w}_*)$ with $\mathbf{W}_* = \mathbf{Y}_* \cup \mathbf{X}_*$, and try to identify it using CTFIDU. Nevertheless, depending on the graphical structure, the original query may be identifiable even if the latter is not. To witness, consider the causal diagram in Fig. 6(a) and the counterfactual $P(Y_x = y \mid Z_x = z, X = x')$, which can be written as $P(Y_x = y, Z_x = z, X = x')/\sum_y P(Y_x = y, Z_x = z, X = x')$. Following the strategy explained so far, the numerator is equal to $P(Y_z = y)P(Z_x = z, X = x')$, where the second ctf-factor is inconsistent, and therefore not identifiable from $\mathbb{Z}$. Nevertheless, the conditional query is identifiable as

$$\frac{P(Y_{xz} = y)P(Z_x = z, X = x')}{P(Z_x = z, X = x')\sum_y P(Y_{xz} = y)} = P(Y_{xz} = y) = P(Y = y \mid Z = z, X = x). \quad (23)$$

To characterize such simplifications of the query, we look at the causal diagram paying special attention to variables after the conditioning bar that are also ancestors of those before. Let $\mathbf{X}_*(W_\mathbf{t}) = \mathbf{V}(\|\mathbf{X}_*\| \cap An(W_\mathbf{t}))$, that is, the primitive variables in $\mathbf{X}_*$ that are also ancestors of $\mathbf{W}_\mathbf{t}$.

**Definition 7** (Ancestral components). *Let $\mathbf{W}_*$ be a set of counterfactual variables, $\mathbf{X}_* \subseteq \mathbf{W}_*$, and $\mathcal{G}$ be a causal diagram. Then the ancestral components induced by $\mathbf{W}_*$, given $\mathbf{X}_*$, are sets $\mathbf{A}_{1*}, \mathbf{A}_{2*}, \ldots$ that form a partition over $An(\mathbf{W}_*)$, made of unions of ancestral the sets $An(W_\mathbf{t})_{\mathcal{G}_{\underline{\mathbf{X}_*(W_\mathbf{t})}}}, W_\mathbf{t} \in \mathbf{W}_*$. Sets $An(W_{1[\mathbf{t}_1]})_{\mathcal{G}_{\underline{\mathbf{X}_*(W_{1[\mathbf{t}_1]})}}}$ and $An(W_{2[\mathbf{t}_2]})_{\mathcal{G}_{\underline{\mathbf{X}_*(W_{2[\mathbf{t}_2]})}}}$ are put together if they are not disjoint or there exists a bidirected arrow in $\mathcal{G}$ connecting variables in those sets.*

**Lemma 3** (Conditional Query Reduction). *Let $\mathbf{Y}_*, \mathbf{X}_*$ be two sets of counterfactual variables and let $\mathbf{D}_*$ be the set of variables in the same ancestral component, given $\mathbf{X}_*$, as any variable in $\mathbf{Y}_*$, then*

$$P(\mathbf{Y}_* = \mathbf{y}_* \mid \mathbf{X}_* = \mathbf{x}_*) = \frac{\sum_{\mathbf{d}_* \setminus (\mathbf{y}_* \cup \mathbf{x}_*)} P(\bigwedge_{D_\mathbf{t} \in \mathbf{D}_*} \mathbf{D}_{\mathbf{pa}_d} = d)}{\sum_{\mathbf{d}_* \setminus \mathbf{x}_*} P(\bigwedge_{D_\mathbf{t} \in \mathbf{D}_*} \mathbf{D}_{\mathbf{pa}_d} = d)}, \quad (24)$$

*where $\mathbf{pa}_d$ is consistent with $\mathbf{t}$ and $\mathbf{d}_*$, for each $D_\mathbf{t} \in \mathbf{D}_*$. Moreover, $P(\mathbf{Y}_* = \mathbf{y}_* \mid \mathbf{X}_* = \mathbf{x}_*)$ is identifiable from $\mathbb{Z}$ if and only if $P(\bigwedge_{D_\mathbf{t} \in \mathbf{D}_*} D_{\mathbf{pa}_d} = d)$ is identifiable from $\mathbb{Z}$.*

Using the notion of ancestral components and Lemma 3, we propose a conditional version of CTFIDU (Algorithm 2). Due to Lemma 3, it is easy to see that CTFID is complete. In case that $\mathbf{W}_* = \mathbf{Y}_* \cup \mathbf{X}_*$ contains nested counterfactuals, Theorem 4 can be used first and the new variables in the expression are added to $\mathbf{W}_*$, then the new sum indexes added to $\mathbf{d}_*$ in Eq. (24).

**Theorem 6** (CTFID completeness). *A counterfactual probability $P(\mathbf{Y}_* = \mathbf{y}_* \mid \mathbf{X}_* = \mathbf{x}_*)$ is identifiable from $\mathbb{Z}$ and $\mathcal{G}$ if and only if CTFID returns an expression for it.*

To illustrate the mechanics of our algorithm, consider the causal diagram $\mathcal{G}$ in Fig. 7(a) and the identification of the query $P(Y_{x_1 Z_{x_0}} = y_1 \mid X_{w_0} = x_1, T_{r_1} = t_1, R = r_0)$ from $\mathbb{Z} = \{\{\}, \{W, T\}\}$. First, we can use Theorem 4 to write the query as:

$$\sum_z P(Y_{x_1 z} = y_1, Z_{x_0} = z \mid X_{w_0} = x_1, T_{r_1} = t_1, R = r_0). \tag{25}$$

The query within the sum can be processed using $\text{CTFID}(\mathbf{Y}_* = \{Y_{x_1 z}, Z_{x_0}\}, \mathbf{y}_* = \{y_1, z\}, \mathbf{X}_* = \{X_{w_0}, T_{r_1}, R\}, \mathbf{x}_* = \{x_1, t_1, r_0\}, \mathbb{Z}, \mathcal{G})$.

At line 1, the algorithm looks for the ancestral components of $\mathbf{Y}_* \cup \mathbf{X}_*$ given $\mathbf{X}_*$. This results in three ancestral components, which can be labeled as $\mathbf{A}_{1*} = \{Y_z, X_{w_0}, T, R\}$, $\mathbf{A}_{2*} = \{Z_{x_0}\}$ and $\mathbf{A}_{3*} = \{T_{r_1}\}$. Then, we let $\mathbf{D}$ be the union of the ancestral sets that contain variables in $\mathbf{Y}_*$, that is $\mathbf{D}_* = \{Y_z, X_{w_0}, T, R, Z_{x_0}\}$. We also gather the values $\mathbf{d}_* = \{y_1, x_1, t, r_0, z\}$, where $t$ is a new value introduced for $T$ that will appear as the index in a sum on line 3.

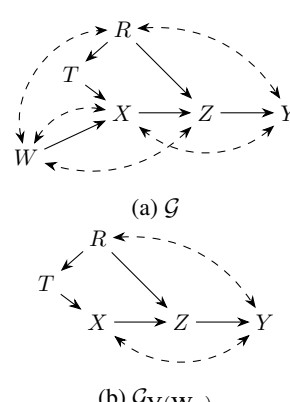

(a) $\mathcal{G}$

(b) $\mathcal{G}_{\mathbf{V}(\mathbf{w}_*)}$

Figure 7: Graphs used for the running example of the algorithms.

At this point, $\text{CTFIDU}$ is invoked with arguments $(\mathbf{Y}_* = \{Y_z, X_{w_0 t}, T_{r_0}, R, Z_{x_0 r_0}\}, \mathbf{y}_* = \mathbf{d}_*, \mathbb{Z}, \mathcal{G})$. In this case lines 1-3 will not make any change. Note that $An(D_{\mathbf{pa}_d}) = \{D_{\mathbf{pa}_d}\}$, hence their union is also an ancestral set, and $\mathbf{W}_*$ will be equal to the $\mathbf{Y}_*$ given to $\text{CTFIDU}$. The c-components in the graph $\mathcal{G}[\mathbf{V}(\mathbf{W}_*)]$ are $\{X, R, Y\}$, $\{T\}$ and $\{Z\}$, then we need to consider the ctf-factors $P(Y_z = y_1, X_{w_0 t} = x_1, R = r_0)$, $P(T_{r_0} = t)$ and $P(Z_{x_0 r_0} = z)$. In this case all of the ctf-factors are consistent, so we can try to identify each of them from $\mathbb{Z}$. Specifically, we will obtain[4]

$$P(Y_z = y_1, X_{w_0 t} = x_1, R = r_0) = P_{w_0, t}(r_0) P_{w_0, t}(x_1 \mid r_0) P_{w_0, t}(y_1 \mid r_0, x_1, z), \tag{26}$$
$$P(T_{r_0} = t) = P(t \mid r_0), \text{and} \tag{27}$$
$$P(Z_{x_0 r_0} = z) = P_{w, t'}(z \mid r_0, x_0); \tag{28}$$

where $w$ and $t'$ could be any values in $\mathfrak{X}_W$ and $\mathfrak{X}_T$, respectively. We could just take $w_0$ and $t$ for simplicity. Then, considering the sum over $z$ introduced for unnesting, the final result is:

$$P(Y_{x_1 Z_{x_0}} = y_1 \mid X_{w_0} = x_1, T_{r_1} = t_1) = \tag{29}$$
$$\sum_{z,t} P_{w_0, t}(r_0) P_{w_0, t}(x_1 \mid r_0) P_{w_0, t}(y_1 \mid r_0, x_1, z) P(t \mid r_0) P_{w_0, t}(z \mid r_0, x_0). \tag{30}$$

## 5 Conclusions

We investigated in this paper the problem of nested and non-nested counterfactual identification from an arbitrary combination of observational and experimental distributions. First, we introduced fundamental building blocks for counterfactual reasoning, which allowed us to prove several properties of nested counterfactual distributions, including the counterfactual unnesting theorem (Theorem 4), and the ancestral and counterfactual factorization theorems (Theorem 1, Theorem 2). Moreover, we introduced a graphical condition (Definition 6, Theorem 3) and developed an efficient algorithm (Algorithm 1) for identifying marginal counterfactuals. We then proved their sufficiency and necessity for the task of nested counterfactual identification (Theorem 5). Lastly, we reduced the identification of conditional counterfactuals to that of marginal ones (Lemma 3) and provided a corresponding complete algorithm (Algorithm 2) for this task (Theorem 6). These results advance the state of the art regarding the distributions the inference engine expects as input, and the query it can generate as the output. In terms of the input, it accepts any combination of observational and experimental distributions, and in terms of the output, it considers an arbitrary nested counterfactual distribution. This work closes the long-standing inferential gap within the layers of the Pearl's Causal Hierarchy, and now one can move within all layers of the hierarchy in a very flexible and general way.

---

[4]For details on this example see Appendix E.

## Acknowledgments and Disclosure of Funding

Elias Bareinboim and Juan D. Correa were supported in part by funding from the NSF (IIS-1704352, IIS-1750807), Amazon, JP Morgan, and The Alfred P. Sloan Foundation. Sanghack Lee was supported by the New Faculty Startup Fund from Seoul National University.

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
