# OpenReview forum: "Nested Counterfactual Identification from Arbitrary Surrogate Experiments"
_NeurIPS.cc/2021/Conference — NeurIPS 2021 Poster_

### Official Review · Reviewer_xPoj · 2021-07-15

**Rating:** 7
**Confidence:** 3

**Summary:**

In this causality paper the evaluation of nested counterfactuals is reduced to that of unnested ones together with an algorithm for identification. This is a dense technical contribution, proofs are clearly detailed in the supplementary material.

**Ethical Concerns:**

Nothing.

**Limitations And Societal Impact:**

Nothing.

**Main Review:**

The contribution is clearly an advance for the state of the art in causality research. I personally missed some attention in better motivating the importance of nested/conditional CFs in practical application, but apart from that I see no reasons for not accepting the paper.

**Time Spent Reviewing:**

2

---

> ### Author Response · Authors · 2021-08-10
> **Response to Reviewer xPoj**
>
> Thanks for pointing out the need for more motivation on nested and conditional counterfactuals. We will make sure to provide some discussion or example related to these types of counterfactuals with the additional page (if the paper is accepted).

---

### Official Review · Reviewer_KaGC · 2021-07-15

**Rating:** 8
**Confidence:** 2

**Summary:**

This submission presents two contributions to the theory of counterfactual inference from observational and experimental data. First, it reduces nested counterfactuals (which are often required for standard estimates in the potential outcomes literature) to their non-nested counterparts. Second, it introduces a necessary and sufficient graphical condition under which these (conditional and marginal / nested and non-nested) counterfactuals are non-parametrically identifiable from observational and experimental data.

**Ethical Concerns:**

None.

**Limitations And Societal Impact:**

The authors have not addressed limitations and societal impact, but because of the theoretical nature this is not necessary.

**Main Review:**

This is overall a very strong submission, tackling an important problem with what appears to be a technically correct approach.

In particular, I found the examples throughout the paper to be particularly helpful in understanding (i) how certain estimands could be connected to nested counterfactuals, and (ii) how the various theorems imply useful conclusions towards identification and estimation. This is particularly important in technically dense theoretical work such as this, and I would even encourage the authors to expand on the existing examples to further improve the paper's readability.

I would appreciate if the authors could respond to a few minor questions about the results.

1. Theorem 5 and Theorem 7 are described as "completeness" results, but the statements of the theorems appear to provide soundness and completeness. Could the authors clarify this?

2. It appears that this submission only provides identification results for counterfactual queries that are averaged over units, and not unit-level counterfactual identifiability results. Could the authors clarify how unit-level identification results may or not differ?

**Time Spent Reviewing:**

3

---

> ### Author Response · Authors · 2021-08-10
> **Response to Reviewer KaGC**
>
> We are thankful for your positive assessment and constructive feedback, refreshing.
>
> We agree that the paper would benefit from having more examples. In addition to more detail in the appendix, we plan to use the additional one page allowed for more examples (if the paper is accepted). Regarding your thoughtful questions, thank you, we will try to address them in the sequel.
>
> Q1. You are right. Theorems 5 and 7 are about soundness and completeness. While soundness follows from the results preceding these theorems, we need to make this clear in the title for the theorems (the statements are fine). Proving the necessity (and therefore completeness) was substantially more challenging (e.g., around pages 16-18 in Appendix C and 20-24 in Appendix D), and we ended up overly excited about it and perhaps ended up highlighting it in the theorem’s titles.
>
> Q2. The input distributions in our identification task describe the average population. Therefore, in general, it is not possible to characterize unit-level counterfactual queries from them. The outcomes observed on average could be very different from those of a particular type of unit. \
> One way to get closer to unit-level counterfactuals is to restrict the analysis to a specific subpopulation by conditioning on some $\mathbf{Z}=\mathbf{z}$. \
> Other approaches such as "Causes of effects: Learning individual responses from population data" (S. Mueller, A. Li, and J. Pearl, ref: arXiv:2104.13730) use further assumptions or bounds on the probability of necessity and sufficiency to achieve some level of individualization. After all, our approach is probabilistic and non-parametric (without making assumptions about the structural functions $\mathcal{F}$ and the probability over the exogenous P(U), such as monotonicity), and  goes as far as providing averages over the unobserved U’s, never the U itself. We acknowledge that there is very interesting work trying to get more individual types of effects to work, including Actual Causality by Pearl, Halpern, and collaborators, as discussed in Ch. 10 in Causality (Pearl, 2000).

---

> > ### Comment · Reviewer_KaGC · 2021-08-31
> > **Thanks to the authors for responding to my minor questions**
> >
> > Thank you for clarifying the scope of Theorem 5 and 7 and the relationship to unit-level counterfactuals.
> >
> > Despite the mixed reviews of this submission, my positive assessment of the work remains unchanged.

---

### Official Review · Reviewer_hjGq · 2021-07-20

**Rating:** 6
**Confidence:** 3

**Summary:**

The authors study the identification of nested counterfactuals from an arbitrary combination of observations and experiments. They prove the counterfactual unnesting theorem, which allows for the mapping of nested to unnested counterfactuals. Then they introduce 1) sufficient and necessary graphical conditions for counterfactual identification from an arbitrary combination of observational and empirical distributions, and 2) an algorithm for identifying nested counterfactuals, the failure of which implies non-identifiability.

**Ethical Concerns:**

None.

**Limitations And Societal Impact:**

The limitations and assumptions of the proposed theoretical work are fully described.

**Main Review:**

The paper is well written and the motivation for the presented theoretical work is well justified. Notation, definitions and assumptions are clearly stated. Though the contributions of the proposed theoretical results have important implications for the analysis, understanding and identifiability of structural causal models, the practical implications beyond of the analytical tools in Algorithms 1 and 2 remain to be seen.

There are few typos beyond those presented in Section F of the SM, thus the paper will benefit from careful proofreading.

**Time Spent Reviewing:**

3.5

---

> ### Author Response · Authors · 2021-08-10
> **Response to Reviewer hjGq**
>
> We thank the reviewer for the positive assessment and the feedback provided, much appreciated. Also, we agree about proofreading and will engage in a deeper reading of the paper to avoid typos, glitches, and other embarrassments.
>
> Regarding the practical implications of our work, we do not disagree with the reviewer but would like to mention that we do see these results as a stepping stone towards having real-world applications that perform counterfactual reasoning in broader settings – with arbitrary causal diagrams, arbitrary nested counterfactuals, and arbitrary combinations of observational and experimental distributions. Historically speaking, we note that the same route happened with interventional queries (without nested counterfactuals or inconsistencies across subscripts, or the antecedent of the counterfactual). The community first had a solid understanding of the more foundational results, and then we have seen an increase in research activity and number of papers applying these methods and considering issues of finite samples, robustness, misspecification, and other very real issues that appear in practice.

---

> > ### Comment · Reviewer_hjGq · 2021-08-23
> > **Discussion**
> >
> > I want to thank the authors and the other reviewers for the thorough discussion. However, after considering the main points of the discussion I have decided to lower my score.

---

> > > ### Author Response · Authors · 2021-08-23
> > > **Re: Discussion**
> > >
> > > Dear Reviewer hjGq,
> > >
> > >  It seemed that we have rebutted every point made by the other reviewers. We consider that the required revision to our paper will be relatively minor, and the revision would not diminish the novelty and contribution of the paper but strengthen the paper.
> > >
> > >  Therefore, we kindly ask whether you can explain why you’ve changed the score? Further, would you please share any remaining concerns about the paper so that we can respond during this discussion phase?
> > >
> > > Sincerely,
> > >
> > > Paper 10135 authors

---

### Official Review · Reviewer_EAGq · 2021-07-31

**Rating:** 5
**Confidence:** 4

**Summary:**

This paper aims to use the theory of causal graphical models to yield a sound and complete identification algorithm for identification of a joint probability of counterfactual events from a combination of observed and interventional distributions.

**Ethical Concerns:**

None noted.

**Limitations And Societal Impact:**

This is a methodological paper causal inference: no potential negative societal impacts are foreseen by this reviewer.

For limitations, please see the main body of the review.

**Main Review:**


The paper is very difficult to follow, as written.  The paper would definitely benefit from worked examples of the operation of the algorithms, and at least an outline of the proof strategy for soundness and completeness.  The proofs are very difficult to follow otherwise.

It appears to generalize results in [10].  However, the paper completely fails to mention ("Complete Identification Methods for the Causal Hierarchy", Shpitser and Pearl, 2008), which gives an algorithm for identification of counterfactual events in terms of all interventional distributions P* -- the same problem the authors consider.

Many of the concepts the authors introduce, e.g. counterfactual factors, eliminating redundancy, and detecting inconsistency of events, and the problem of identifying a joint probabilities of counterfactual events and conditional probabilities of counterfactual events (at the third level of the ladder of causation) in terms of interventions distributions (at the second level) first appeared in the above reference.
Specifically, identification of counterfactual joints and conditionals is given by the ID* and IDC* algorithms, respectively.

In addition, the above reference gives a graphical view of the problem based on counterfactual graphs (a generalization of Balke and Pearl's twin network graphs) that shed light on many concepts the authors use that are typically graphical (e.g. ancestors of counterfactuals -- in what graph?, counterfactual factors -- corresponding to districts in what graph?, ancestral set factorization -- with respect to what graph?) but appear not to be in the authors' paper.

Given that the omitted paper deals with exactly the same problem, the omission is quite disappointing, and leads to serious worries regarding novelty, and the true relationship of the authors' work to prior work.  Are the authors' algorithms effectively the same as ID* or IDC*?  Did these algorithm "miss something" that the authors caught?

Other comments:

Theorem 1: this appears to be definition of notation, rather than a theorem.  In other words, the semantics the authors use give the definition of the joint distribution over counterfactuals using the structural model (via equation (1)), while Theorem 1 is a definition of the "capital letters in the subscript" notation, which is otherwise undefined via (1).

Minor comments:

"Further, there are also result on the identification of path-specific effects, which correspond to counterfactuals that isolate specific paths in the graph. In particular, [1] provides a complete algorithm for identification from observational data."

Unlike other results the authors cite, [1] appears to only apply to fully observed models.  The extensions to hidden variable models appear in ("Identification of Personalized Effects Associated With Causal Pathways" Shpitser and Sherman, 2018).

---

"the composition axiom [13, 7.3.1]"

To most readers, this is probably known as consistency.  It might be worth pointing this out to improve readability.


**Time Spent Reviewing:**

4 hours.

---

> ### Author Response · Authors · 2021-08-10
> **Response to Reviewer EAGq**
>
> Thanks for reviewing our paper. We will address the points you raised one by one.
>
> ### Need for examples and proof outlines
> Although we provided a non-trivial running example of the algorithm in Appendix E, we will include some more in the main body of the manuscript using the extra page (if the paper is accepted). This is a good suggestion, thank you. We will also add brief and informative outlines for the proof of the main results.
>
> ### “Failure” to mention Shpitser and Pearl 2018
> >the paper completely fails to mention ("Complete Identification Methods for the Causal Hierarchy", Shpitser and Pearl, 2008)
>
> The article "Complete Identification Methods for the Causal Hierarchy" (Shpitser and Pearl, 2008) (SP08) consolidates several results on identification of causal effects. The portion related to our work (Section 4 in SP08) was first introduced in “What counterfactuals can be tested” (Shpitser and Pearl 2007), which we cited as [17] and discussed in lines 58-60 in the submitted paper. So, there is no intent of avoiding citation, we think it’s only fair to cite the first paper that handles the problem and not the more general survey. If you feel there are some substantive results related to our submission that were not present in the original paper from 2007, we would be happy to reconsider the citation.
>
> >which gives an algorithm for identification of counterfactual events in terms of all interventional distributions P* -- the same problem the authors consider
>
> Furthermore, we would like to highlight the differences between the problem setting (and results) in those papers and ours, as we already listed in lines 58-60 of the submitted paper. First, while they assume $P_*$ -- the set of all possible interventional distributions -- is available, our method allows for a user-specified subset of $P_*$ corresponding to observations and experiments available to the analyst. In other words, a major assumption that the observational distribution and all combinations of experimental distributions are available is relaxed.  Second, their work does not consider or give any formal account for queries involving nested counterfactuals, as discussed in the summary of our contributions (lines 78-83). This class of counterfactuals is considered significant in the literature and was highlighted in our title. Third, their work does not include any time complexity analysis or efficiency guarantee. Moreover, while proving completeness of ID*, both SP08 and [17] seem to have missed some non-identifiable instances corresponding to case i in our proof of Lemma 5 (line 511-537 in Appendix C).
> We take the confusion on this point as a word of caution and will make sure these differences are prominently highlighted in our paper. We hope the reviewer will improve his/her assessment as there is no omission of this reference and the relevant works were acknowledged and cited in lines 50-68 of the submitted manuscript.
>
> ### Graphical view of the problem
> We define Counterfactual ancestor, Ancestral set factorization, Counterfactual factorization, and similar notions with respect to the original causal graph $\mathcal{G}$. Employing the original graph (and simple transformations of $\mathcal{G}$) for those concepts avoids some shortcomings of counterfactual graphs, including:
> - Size. The construction of a counterfactual graph includes one subgraph of the original diagram for each different submodel (i.e., interventions) involved in the query. Moreover, counterfactual graphs have additional edges to make unobservable variables shared across all subgraphs. In terms of readability, even instances with 4 or 5 variables are impractical to draw by hand or inspect by eye.
> - Need for Redundancy Control. After the initial construction described above, Shpitser &  Pearl’s method checks every pair of nodes corresponding to the same original variable for two conditions.
>
> The notions of counterfactual ancestors and ancestral factorization make our method impervious to these issues. We further note that twin networks, counterfactual graphs, and other graphical structures are derived data-structures from the original causal diagram; they are not primitives in the structural semantics. The route discussed above takes this primitive, the causal diagram, and bypasses the construction of intermediate data-structures to obtain the constraints necessary for the specific counterfactual identification task discussed in this paper.
>
>
> ### About reference [1]
> >Unlike other results the authors cite, [1] appears to only apply to fully observed models. The extensions to hidden variable models appear in ("Identification of Personalized Effects Associated With Causal Pathways" Shpitser and Sherman, 2018).
>
> Thanks for bringing attention to this. This citation was meant to be Shpitser and Sherman, 2018. After submission, we realized this mistake and acknowledged it in the supplemental material together with other minor typos, as discussed in lines 761-784.
>
> ### About Theorem 1
> > this appears to be definition of notation, rather than a theorem
>
> Eq. (3) gives a mapping from an arbitrary nested counterfactual to another with one level of nesting less. This mapping is justified from the semantics of SCMs and the definition of “derived model” and “potential response” provided before in the paper. This formally grounds the relationship between nested counterfactuals and SCMs, which as far as we know, has not been clearly stated elsewhere.
>
> While it is true that Eq. (3) could be taken as an axiom, it is more transparent to present it as a consequence of the semantics of the framework. This is similar to the choice of taking the composition, effectiveness, and recursiveness as axioms (Pearl 2000, 7.1.3) or as consequences of the structural semantics. It depends on what one wants to stipulate and which aspects of the causal puzzle one wants to investigate.

---

> > ### Comment · Reviewer_EAGq · 2021-08-13
> > **re: response.**
> >
> > re: "“Failure” to mention Shpitser and Pearl 2018":
> >
> > Fair enough -- I think more folks are familiar with the JMLR reference.
> >
> > The amount of space you devote to ID*/IDC* appears to be limited to the following sentence:
> >
> > "Moreover, [17] studied the identification of arbitrary (non-nested) counterfactuals under the assumption that data from experiments in every variable is available. Yet, the problem of identifying such quantities from a subset of the space of all experiments remains open."
> >
> > This is despite the fact that there is a _significant_ conceptual overlap of tools and ideas used.
> >
> > re: differences from ID*/IDC*:
> >
> > I think discussing these differences in detail clearly, as you have aimed to do in the response above, is what seems to be missing in the submitted draft, and what made the evaluation of novelty more challenging than it needed to be.
> >
> > "their work does not include any time complexity analysis or efficiency guarantee."
> >
> > Perhaps I am missing something, but does this paper include this?
> >
> > re: redundancy control/counterfactual graphs:
> >
> > Presumably, redundancy control is necessary because of consistency, not because graphs are used.
> >
> > It sounds like you claim that ID* does "too much work" when checking for redundancy, and your proposed algorithm cuts down on this.  This is precisely the kind of thing that would have been valuable to discuss.
> >
> > re: nested counterfactuals:
> >
> > Perhaps I am missing something, but you assume the query is always a joint (or conditional) of counterfactual events, and point out (via CUT) that a nested counterfactual is a function of such a joint (botton of p. 4).  This is true, but when we nest, we compute a margin, and sometimes margins are identified when joints are not.  But it doesn't seem as if you give a completeness argument for the nested counterfactual margins, only for the unrolled joint.  Is that correct?
> >
> > Minor comments (these are orthogonal to the evaluation of the paper):
> >
> > I hope it is fair to assume that the authors find Pearl's graphical approach to causality valuable.  If so, their position seems to be a somewhat curious one: "please use graphs, but only if the problem is not too complicated, otherwise fall back on algebra."
> >
> > Ignoring the problem of counterfactual event identification for the moment, graphs arising in more elementary problems, such as identification of causal effects can be very complicated indeed.  Presumably, this is merely a reflection of the fact that some structured systems graphs represent are complicated.  Surely, if someone finds graphs valuable, it would be a strategic mistake to abandon graphs in such problems -- as they make a complicated problem simpler, not harder.
> >
> > re: CUT:
> >
> > It seems that counterfactuals that use capital letters in the subscript are simply not defined without CUT.  So it's not clear what CUT is saying unless it's viewed as a definition.

---

> > > ### Author Response · Authors · 2021-08-15
> > > **follow up**
> > >
> > > Thank you for reading our response and following up. We are glad it has become clear that this important reference was indeed cited and discussed. It seems the discussion now is regarding how much detail and depth is appropriate. Please see below our considerations regarding this issue.
> > >
> > > >The amount of space you devote to ID*/IDC* appears to be limited to the following …
> > > >
> > > >This is despite the fact that there is a significant conceptual overlap of tools and ideas used.
> > > >
> > > >I think discussing these differences in detail clearly, as you have aimed to do in the response above, is what seems to be missing in the submitted draft, and what made the evaluation of novelty more challenging than it needed to be.
> > >
> > > It seems the reviewer agrees that our method, and more importantly, our task are different from those discussed by SP08, [17]. We are not opposed to improving the legibility of our paper, as already stated in our initial response, and we can elaborate on the differences between the tasks. Having acknowledged that the tasks are different, an explicit discussion on the differences between the methods to solve them does not seem essential, and maybe distracting or confusing for most potential readers interested in the task solved in our paper. Abstractly, if a paper solves a task `T1` with method `M1`, and another paper solves a task `T2` with method `M2`, it’s not necessarily the case that the authors need to use, or even justify why another method `M1` can or cannot be used to solve `T2`. Still, after reading the review, we acknowledge that some people may be comfortable with `M1` and may want to try to build on it to solve `T2`; as we already mentioned in the rebuttal, this was not the route taken by this paper. More critically, most (if not all) of the tools at the intersection of [17]’s and our method such as c-components, c-component factorization, extending non-identifiability to descendants in the graph, etc that have been previously defined in the literature were properly acknowledged in our paper. This includes the fundamental results by Tian & Pearl [22, 23] and Huang & Valtorta [7, 8].
> > >
> > > We believe this specific comparison to ID* & IDC* is perhaps a discussion appropriate for a survey paper on counterfactual identification. The question you asked about complexity analysis is one good example of this. We have not cited it in our paper since our goal was not to be critical about SP08, which solved a different task than ours, or to survey the literature in a deep way, noting that this is a conference paper solving a different task. Nevertheless, we would be glad to provide a brief discussion in the appendix for readers interested in this aspect, assuming we understood the reviewer’s suggestion properly.
> > >
> > > > Presumably, redundancy control is necessary because of consistency, not because graphs are used.
> > > >
> > > > It sounds like you claim that ID* does "too much work" when checking for redundancy, and your proposed algorithm cuts down on this. This is precisely the kind of thing that would have been valuable to discuss.
> > >
> > > Again, our goal is not to be critical about SP08’s method or to analyze their contributions in detail, as it lies beyond the scope of the task and contributions provided in our paper. Still, you are right, technically speaking, redundancy control is needed because of consistency. While SP08 first constructs a counterfactual graph and later finds redundant nodes, our method guarantees consistency by forming ancestral sets and counterfactual factors, which together with lemma 1, naturally avoid redundancy before it becomes an issue. This means that if we were to constrain our task to match the one solved by SP08, our method could also be a solution. To be clear, we are engaging in this discussion and somewhat detailed comparison for the sake of addressing the reviewer’s questions. It is beyond the contribution of this paper to survey or criticize previous results that do not solve the same task.
> > >
> > >
> > > > Minor comments (these are orthogonal to the evaluation of the paper):
> > > >
> > > >I hope it is fair to assume that the authors find Pearl's graphical approach to causality valuable. If so, their position seems to be a somewhat curious one: "please use graphs, but only if the problem is not too complicated, otherwise fall back on algebra."
> > > >
> > > > Ignoring the problem of counterfactual event identification for the moment, graphs arising in more elementary problems, such as identification of causal effects can be very complicated indeed. Presumably, this is merely a reflection of the fact that some structured systems graphs represent are complicated. Surely, if someone finds graphs valuable, it would be a strategic mistake to abandon graphs in such problems -- as they make a complicated problem simpler, not harder.
> > >
> > > We appreciate the opportunity to clarify this issue, which is substantive, and will try to take it as an “intellectual exchange of ideas.” Even though this is posed as a minor comment and “orthogonal to the evaluation of the paper,” there are misconceptions about our work and illations about our standpoint that are undeserved.
> > >
> > > First of all, we certainly find the graphical models approach to causality valuable, where Pearl is one of the pioneers and leaders. Technically speaking, this is why the main concepts and results in our paper, including counterfactual ancestors, ancestral set factorization, and counterfactual factorization, are all defined with respect to the causal diagram $\mathcal{G}$. In addition to what we already said in the paper and our rebuttal, the causal diagram is *the* primitive, the primary object that encodes the necessary assumptions for performing the task investigated in our paper.
> > >
> > > Having acknowledged the centrality of causal diagrams for our analysis, this is not to say that any data structure derived from the causal diagram cannot be more complicated than necessary or exhibit different properties that are more or less interesting depending on the context. Let us make it clear that we are not against the causal diagram at all. In our previous response, we were simply stating that we didn’t take the data structure “counterfactual graph” as input, which is derived from the causal diagram itself, but it’s not the same. Therefore, there is no “abandonment” of the causal diagram at all and at any point of our method; on the contrary, it plays a central role in the whole process. We are not sure if you are raising an issue regarding ignoring the counterfactual graph defined in SP08, which we think was addressed above. Philosophically speaking, we consider it a valid approach to take a causal diagram as input and develop any convenient data structure suitable for a specific task. The causal diagram is a transparent and convenient construction for eliciting background knowledge, but the data structure used for intermediate steps inside an algorithm based on this graph can be whatever is more convenient and has the desired properties for solving a certain task, obviously, assuming they preserve the validity of the inferences.
> > >
> > > >It seems that counterfactuals that use capital letters in the subscript are simply not defined without CUT. So it's not clear what CUT is saying unless it's viewed as a definition.
> > >
> > > This is not accurate. Definition 1, 2, and Eq. (1) are sufficient to define counterfactuals “that use capital letters in the subscript,” that is, nested counterfactuals. As mentioned in our response, the CUT does not define nested counterfactuals, it connects nested counterfactuals with unnested ones. This connection follows from the semantics of the framework, it is not taken as a definition or axiom.

---

### Official Review · Reviewer_mQGH · 2021-08-20

**Rating:** 4
**Confidence:** 3

**Summary:**

The paper presents identification algorithms for counterfactuals  and conditional counterfactuals. The counterfactuals can be nested. The input consists of observations and experiments were the interventions can be arbitrary. Proofs for soundness and completeness are provided. No software is included.

**Limitations And Societal Impact:**

 I would say that "Arbitrary surrogate experiments" in the title is a too strong statement. Although the interventions can be arbitrary, all variables are always measured in the experiments.

**Main Review:**

[Note: I am an additional reviewer and have not seen the earlier comments or the authors' rebuttal.]

This is a potentially strong paper but I would not like to see it published in the current form because I see that the presentation could be improved in many ways. I am afraid that publishing the paper in the current form would create confusion. On the other hand, I think it is relatively easy to fix the paper and resubmit.

1) The authors do not explain how their proposal differs from the current state-of-the-art.  For years I have believed that the algorithm published by Shpitser & Pearl (2007,2008) completely solves the counterfactual identification problem. The manuscript implicitly indicates that this is not the case but do not present the novel contribution in clear and direct way. It is should not be readers' task to find the subtle differences between the proposed algorithm and the algorithm published by Shpitser & Pearl. I would like to see the differences explicitly stated and demonstrated in an identification problem that is solvable with the proposed algorithm but not with the existing algorithms.

2) The area chair has expressed a concern that a query could be identifiable from marginalization of  some other form than the one considered (e.g. the last line of Algorithm 1). I share this concern because I cannot see an immediate reason why some other marginalization could not be used.

3) The  notation gets really heavy in some places like in line 9 of Algorithm 1.

4) I traced algorithm 2 in Fig. 5(a) and obtained the same result as in equation (23). I ran a simulation to check the result numerically. So I can confirm that the algorithm works correctly in this example.  Proofs are of course needed but these kind demonstrations could serve the average reader who does not have the perseverance to study the proofs in detail.

**Time Spent Reviewing:**

10

---

> ### Author Response · Authors · 2021-08-23
> **Response to Reviewer mQGH**
>
> We thank the reviewer for taking time to provide an additional review to our paper.
>
> >The authors do not explain how their proposal differs from the current state-of-the-art. For years I have believed that the algorithm published by Shpitser & Pearl (2007,2008) completely solves the counterfactual identification problem.
>
> It depends on how the problem of “counterfactual identification” is defined. If it means in the sense of having all interventional distributions available as input or unnested counterfactuals as the target query, yes, it was solved by Shpitser & Pearl (07,08) (SP0708). Acknowledging that this is an important result, we note it’s a somewhat stringent requirement to say that all interventional distributions are available. We studied a different identification problem following the discussion below, and as explained throughout our paper.
>
> >The manuscript implicitly indicates that this is not the case but do not present the novel contribution in clear and direct way. It is should not be readers' task to find the subtle differences between the proposed algorithm and the algorithm published by Shpitser & Pearl. I would like to see the differences explicitly stated and demonstrated in an identification problem that is solvable with the proposed algorithm but not with the existing algorithms.
>
> We respectfully disagree with the reviewer on the differences being “subtle.” We want to emphasize that the task in our paper is not the same as the one addressed in SP0708, both the input and the output of the task are different. In particular, there are fundamental differences between assuming that ALL interventional distributions are known and allowing for an ARBITRARY subset of user-specified observational and experimental distributions. For instance, does the treatment provided in SP0708 entail sufficient and necessary conditions for identifying counterfactuals from observational data? As far as we can tell, SP0708 does not make any claim in this case, whereas our method is sufficient and necessary for this task by specifying the input as just the empty experiment ($\mathbb{Z}=\{\emptyset\}$). Thus, even at this baseline scenario, our results already answer a question SP0708 cannot. Moreover, adding arbitrary experimental distributions to the input only increases the gap between these works. Finally, we consider nested counterfactuals queries while SP0708 considered unnested ones. Again, SP0708 makes no claim about this scenario.
>
> As it was published, SP0708’s method can only compare with ours for the subset of tasks that SP0708 considers, for which we already know it is complete, making the comparison pointless in terms of decidability. In fact, we are not trying to claim or get credit for anything they claimed; despite the relationship between the problems, our solution has different properties. Our goal is not to survey or criticize this particular paper, we are simply solving another task. (Please, refer also to the answer to reviewer EAGq.)
>
> In terms of the methods, it may be the case that ID* and IDC* could be extended or used within a larger framework that connects their output with the actually available distributions. Nevertheless, adapting those algorithms or proving their completeness for a new problem is not a must. Is it suggested that solving a new problem requires either adapting existing tools (developed to solve a different problem) or formally showing how they cannot be adapted?  We hope the reviewers agree that developing new tools for a problem the current literature cannot handle takes nothing away from the relevance, novelty, or correctness of our results.
>
> Finally, as we discussed in our response to reviewer EAGq, we will make sure to expand on lines 58-60 in the paper to make the differences between the results in SP0708 and our results clear to the reader.
>
> >The area chair has expressed a concern that a query could be identifiable from marginalization of some other form than the one considered (e.g. the last line of Algorithm 1). I share this concern because I cannot see an immediate reason why some other marginalization could not be used.
>
> We agree with the AC and the reviewer on this observation: it is possible, in general, for the query to be identifiable in a form other than the one in the last line of Algorithm 1.  Nevertheless, we are unsure about the concern shared by the area chair and the reviewer. It is proven in Lemma 2 the equivalence between the identification of the original query involving $\mathbf{Y}$*
>  and that of a query with its ancestral set $\mathbf{W}$*. Similar results can be found in most identification results with completeness, including [3,8,9,10,16,21].
>
> However, it is true that the $\mathbf{W}$* may not be minimal in a sense that there can be a subset of $\mathbf{W}$* that can be used to yield an identification formula (which might be compact). Still, this is orthogonal to the identifiability itself as it should be concerning only in the absence of completeness, which is not the case here. If one wants to compute such $\mathbf{W}’_* \subset \mathbf{W}_*$, one can subsequently marginalize out a variable not involved in the query from $\mathcal{G}$ by checking the result of ctfID with $\mathbb{Z}$ refined and the projection of $\mathcal{G}$.
>
> >The notation gets really heavy in some places like in line 9 of Algorithm 1.
>
> We apologize for the confusion here. If the issue is with the bracket notation, line 9 declares that the left hand side probability is equivalent to the evaluation of the identification formula returned by *Identify*. We can change the bracket subscript to using a big vertical bar, which is more standard. Note that the actual evaluation is performed at the end of the algorithm (e.g., lazy evaluation). If the issue is related to the subscript itself, we will think about it and also consider suggestions for a friendlier way of writing this. After all, even though the mathematical treatment provided in the paper is somewhat sophisticated and non-trivial, we are committed to improving and trying to make its content more accessible.
>
> > I traced algorithm 2 in Fig. 5(a) and obtained the same result as in equation (23). I ran a simulation to check the result numerically. So I can confirm that the algorithm works correctly in this example. Proofs are of course needed but these kind demonstrations could serve the average reader who does not have the perseverance to study the proofs in detail.
>
> Thanks for taking the time to go through the algorithm and even verifying an instance numerically. We would be glad to add a demonstration like this in the appendix. Your interest in the paper gives us hope that you can see its value and that our goal is not to avoid recognizing prior work.
>
> >I would say that "Arbitrary surrogate experiments" in the title is a too strong statement. Although the interventions can be arbitrary, all variables are always measured in the experiments.
>
> For the sake of context, we would like to mention that other works published in the literature use the construction "arbitrary surrogate experiments" to represent similar input sets (e.g., [10]), so our title is not without precedent. After all, we believe this is a subjective matter. Still, is there any particular suggestion the reviewer would like to make?

---

> > ### Comment · Reviewer_mQGH · 2021-08-24
> > **Differences with Shpitser & Pearl (2007,2008)**
> >
> > I agree that there are differences between the algorithms but pointed out that the paper does not make them explicit to the reader.
> >
> > All experiments vs a subset of experiments may seem as a major difference between the algorithms but the issue is not straightforward. The algorithm by SP0708 takes a query of third ladder as an input and returns a query of second ladder (or fail) as an output. It instantly comes in the mind that one could apply the algorithm by Lee, Correa & Bareinboim (2019) to the output of the SP0708 algorithm to check the identifiability when only a subset of experiments is available. This would not require developing new methods. I hope this explains why I suggested that the differences between the algorithms should be studied carefully.
> >
> > I suggest replacing "arbitrary surrogate experiments" by "surrogate experiments with arbitrary interventions"

---

> > > ### Author Response · Authors · 2021-08-26
> > > **Re: Differences with Shpitser & Pearl (2007,2008)**
> > >
> > > Since this is an interesting point for the reviewers and potential readers who are familiar with SP07, we have spent some time looking into the properties of an algorithm that runs **gID** (Lee, Correa & Bareinboim, 2019) in the last step of **ID*** (SP07), which we will call **gID***. For the sake of context, **gID*** could be a candidate to replace Algorithm 1 on page 8.  **gID*** soundness is obvious, of course, but the completeness is considerably more challenging as in most of the proofs related to causal identification. In other words, it’s possible that the chaining of these two algorithms could lead to an incomplete solution, and this would, for us, be a pretty weak contribution. (This basic result (soundness) could indeed have been immediately stated as a corollary in (Lee, Correa, Bareinboim, 2019). )
> > >
> > > More interestingly, we believe it can be shown that the failure of **gID*** implies the existence of a ctf-factor $(\mathbf{C}{\*}= \mathbf{c}{\*})$ corresponding to a c-component of $\mathcal{G}[\mathbf{V}(An({\mathbf{Y}\*)})]$ that is either inconsistent or not identifiable by **gID** from $\mathbb{Z}$ and $\mathcal{G}$. Still, the necessity of **gID*** could be shown as a corollary of the completeness of **ctfID**. Specifically, we can use Theorem 2-4, Lemma 2-3, and Lemma 5 similarly to the completeness proof of **ctfID** (Theorem 5, Algorithm 1). An important aspect of these results is that they provide a systematic way of deriving counterexamples that agree on all the distributions given in $\mathbb{Z}$ while disagreeing in the counterfactual query. Moreover, as with our proposed **ctfID**, Theorem 1 of the paper can be used as a pre-processing step to perform the unnesting and solve nested counterfactual queries.
> > >
> > > We already added as a corollary of our paper a formal statement showing that **gID*** is complete, noting that this result follows from the new machinery developed to prove the completeness of **ctfID** (including the new counterfactual factors, counterfactual factorization, the strategy to construct counter-examples, and other results in lemma 2, 3 and theorem 4). We apparently cannot upload a file to the system but can share a link in case the reviewers want to check the proof of this statement.
> > >
> > > After all, our paper solves a significant problem and the technical results are novel and highly non-trivial. We hope the connection provided above can clear the concerns of reviewers mQGH & EAGq, now that the characterization of a modified version of **ID*** & **gID** has been formally established. Importantly, we reiterate that we do not believe this result is indeed required since these other papers tackle different challenges. Still, after interacting with the reviewers, we agree that this is a natural question for a certain portion of the community. Further, we cannot make at this point a precise statement regarding the complexity of this new algorithm but hope that its explicit connection and the sufficiency of **gID*** satisfy the reviewers.

---

### Decision · Program_Chairs · 2021-09-27

**Decision:**

Accept (Poster)

**Comment:**

This paper was discussed at length between the AC and SAC, and between the SAC and program chairs.  The AC was strongly against publication due to the lack of appropriate engagement with prior work.  The SAC and program chairs agree with this concern.  The discussion of the relationship with prior work is insufficient and absolutely must be addressed in the final version.  However, in light of the positive sentiment from four out of five reviewers, we believe the paper should be accepted.

(We are counting as "positive sentiment" one of the expert reviewers who gave the paper a low score, based on their comment: "This is a potentially strong paper but I would not like to see it published in the current form because I see that the presentation could be improved in many ways. I am afraid that publishing the paper in the current form would create confusion. On the other hand, I think it is relatively easy to fix the paper and resubmit.")

We expect the authors to address the points raised by the expert reviewers.  We would not be accepting this paper if we didn't think this was something the authors were capable of doing.

The original meta-review provided by the AC follows.

----

A summary from one of the reviews:

"The authors study the identification of nested counterfactuals from an arbitrary combination of observations and experiments. They prove the counterfactual unnesting theorem (CUT), which allows for the mapping of nested to unnested counterfactuals. Then they introduce 1) sufficient and necessary graphical conditions for counterfactual identification from an arbitrary combination of observational and empirical distributions, and 2) an algorithm for identifying nested counterfactuals, the failure of which implies non-identifiability."

The paper was initially received fairly positively. However, some positive reviews were very short and lacking detail, and all were relatively low confidence. As a result, an additional expert reviewer's opinion was solicited, and (at the behest of the senior AE), another expert reviewer.

The paper received an extensive discussion, both with the authors and among reviewers. The result of this discussion was an overall reduction of scores. The concerns were as follows.

First, the authors' contribution builds on prior work on identification of joints and conditionals of counterfactual events. While the authors do cite this work, both expert reviewers independently felt that there was far from a sufficient comparison to it. In particular, while it is clear that the authors consider a generalization (where only some interventional distributions are allowed, in the spirit of (Lee et al, 2019)), it is not at all clear if sufficiently novel methods, in light of existing prior work, are needed to address this generalization. The authors' manuscript, as it is currently written, does not make it clear what is novel developments unique to the paper, and what is a reformulation of known prior work.

Second, the authors claim to consider identification of nested distributions, and discussion of nested distributions forms much of the paper (and indeed appears in the title). However, the authors do not consider a complete algorithm for the problem of identifiability of nested counterfactuals (and indeed did not seem to answer a direct question about this). Thus, reviewers felt the contribution, as written, did not properly emphasize the actual target they considered. In other words, it is true that the algorithm considered nested counterfactuals as valid targets, but only in a weak sense that such a counterfactual is a (marginal) function of a joint set of events (via what the authors call the CUT). But it is surely the case that in some cases marginals are identified while joints are not.

As a result, the reviewers felt the paper needs a restructure and another round of reviews.

Feedback for authors.

Two independent expert reviewers agreed that the appear needs a more extensive discussion of prior work, in order to make it clearer how novel the contribution is. The authors also need to make it clearer what the completeness argument applies to. It appears not to apply to nested distributions (a class of queries appearing in the title). This is quite confusing. Please address these issues when preparing your revision.